# Analysis Behavior of Openings on Full-Size Cross-Laminated Timber (CLT) Frame Shear Walls Tested Monotonically

**Rudi Dungani [1,*], Sulistyono [2], Tati Karliati [1], Yoyo Suhaya [1], Jamaludin Malik [3], Alpian [4] and Wahyu Supriyati [4]**

[1]  School of Life Sciences and Technology, Institut Teknologi Bandung, Bandung 40132, Indonesia
[2]  Faculty of Forestry, Kuningan University, Kuningan 45513, Indonesia
[3]  Research Center for Biomass and Bioproducts, National Research and Innovation Agency, Bogor 16610, Indonesia
[4]  Department of Forestry, Faculty of Agricultural, Palangka Raya University, Palangka Raya 73112, Indonesia
[*]  Correspondence: dunganir@gmail.com

**Abstract:** Walls, as components of the lateral-force-resisting system of a building, are defined as shear walls. This study aims to determine the behavior of shear wall panel cross-laminated-timber-based mangium wood (*Acacia mangium* Willd) (CLT-mangium) in earthquake-resistant prefabricated houses. The earthquake performance of CLT mangium frame shear walls panels has been studied using monotonic tests. The shear walls were constructed using CLT-mangium measuring 2400 mm × 1200 mm × 68 mm with various design patterns (straight sheathing, diagonal sheathing/45°, windowed shear wall with diagonal pattern and a door shear wall with a diagonal pattern). Shear wall testing was carried out using a racking test, and seismic force calculations were obtained using static equivalent earthquake analysis. CLT-mangium sheathing installed horizontally (straight sheathing) is relatively weak compared to the diagonal sheathing, but it is easier and more flexible to manufacture. The diagonal sheathing type is stronger and stiffer because it has triangulation properties, such as truss properties, but is more complicated to manufacture (less flexible). The type A design is suitable for low-intensity zones (2), and types B, D, E1 and E2 are suitable for moderate-intensity zones (3, 4), and type C is suitable for severe-intensity zones (5).

**Keywords:** *Acacia mangium*; shear wall; monotonic test; seismic resistance; wood-frame

## 1. Introduction

Indonesia is an area prone to disasters and earthquakes, and there are many victims due to construction failures. It is necessary to build a livable house that is easy and fast to build, affordable, with materials available on-site, that is easy to make and meets the requirements for a residential house in the form of a prefabricated house. Some of the benefits of building prefab houses are the fast construction time because they use industrial fabricated modules, a cleaner construction environment and more affordable costs [1].

Earthquakes cause lateral forces on buildings that are random and cyclic [2], depending on the type of ground motions and the characteristics of the building structure [3,4]. Furthermore, in Indonesia, the most seismically active country [5], 85% of the residences are of a wood-framed construction [6]. Shear walls are the vertical, lateral force-resisting element in light-frame constructions, and in 2010, over 90% of Indonesian wooden buildings used them as the primary lateral load-resisting system. These structural components perform very well in high winds and seismic-prone zones [7].

Many attempts have been made by several researchers to investigate the shear resistance of structures to prevent wall sliding by improving the design of shear walls. Di Gangi et al. [8] designed light-frame bamboo shear walls with various height-to-length ratios and these walls were attached to the foundation by means of hold-down anchors and shear bolts. They reported that the performance of the segmented walls did not depend on the aspect ratio after monotonic tests. They also observed that the shear strength of the narrow walls did not degrade at high deflections due to a small displacement demand on the sheathing-to-framing connections. Meanwhile, Di and Zou [9] investigated light

wood frame structures as sheathing wall materials, and the frame shear walls were tested monotonically. They reported that the joints have a high bearing capacity and stiffness during the initial loading stage.

Shear walls of various sheathing also show enhanced performance in one or more areas. The configurations of light timber walls sheathed with one or two sides have increased ductility, stiffness and strength [10,11]. Clech et al. [12] also looked at the effects of the assemblies, particularly on the corners where two walls meet. Further, there are other variations, including the openings in constructing shear walls. Studies on wooden houses structurally designed by utilizing shear walls with and without openings made of wooden frames and various sheathing materials have been undertaken in the past by many researchers. For carbon fiber-reinforced polymers, Husin et al. [13] tested various types of shear walls with various heights, widths and openings under monotonic and cyclic loads. Demirkiran et al. [14] suggested that the position of the opening does not have an effect on the racking resistance of shear walls. It is further reported that the racking resistance of shear walls decreases as the size of the opening increases. Shear walls with various openings and height–width ratios were also tested using the pseudo-dynamic method in which the loading sequences adopted from actual earthquakes were applied to shear wall specimens [15].

Wood products, such as laminated veneer lumber (LVL) and cross-laminated timber (CLT), are new mass products that are widely used in connectors [16,17], ductile connections [18], and pre-fabrication [19]. CLT is an innovative laminated wood product in the form of a quasi-rigid composite that is an engineered wood product, such as plates, which usually consists of an odd number of layers (usually three, five or seven layers), each made of planks placed side by side, which are arranged transversely to each other at an angle of 90°, and they are able to withstand loads entering and leaving the plane [20]. Gagnon and Pirvu [21] suggested that cross-lamination itself provides improved dimensional stability and thermal insulation and a fairly good response in the case of fire, which are added benefits resulting from CLT's massiveness. Furthermore, CLT is a clean product to work with, resulting in little waste or dust produced on-site, which is preferable in terms of health and safety.

The performance of CLT-mangium-frame shear walls was tested using various design patterns (straight sheathing, diagonal sheathing/45°, windowed shear wall with diagonal pattern and door shear wall with diagonal pattern) under monotonic tests. This study aims to determine (1) the effects of openings on full-size CLT-mangium frame shear walls tested monotonically (2) and calculate seismic forces to determine the appropriate earthquake zone.

## 2. Materials and Methods

### 2.1. Preparation of Cross-Laminated Timber (CLT)

A three-ply CLT as a component of frame shear walls was made with mangium (*Acacia mangium* Wild) wood for the face, core and back layers. Mangium wood can be classified as low density, with a density of 0.51 g·cm$^{-3}$. The dimensions of the face and back layer components were 1200 mm by 140 mm by 5 mm (length × width × thickness), respectively. These layers consisted of one panel, while the core layer consisted of smaller parts with dimensions of 140 mm by 20 mm by 5 mm (length × width × thickness), respectively. The fiber orientation of the core layer was perpendicular to the face and back layers.

In the manufacturing of CLT, phenol–formaldehyde (PF) adhesive is spread on the face and back layers with an amount equal to a 170 g·m$^{-2}$ single glue line, followed by pressing at 1.1 MPa for 3 h at room temperature; conditioning took 10 days (Figure 1). Five replications were made for the mangium wood specimens, and all of the CLT-mangium was made in a Wood Laboratory, Institut Teknologi Bandung, Indonesia.

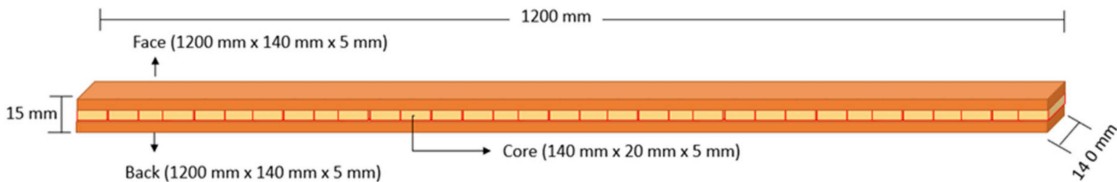

**Figure 1.** Scheme of cross-laminated timber (CLT) manufacturing.

### 2.2. CLT-Mangium Design as a Shear Wall Component

Shear wall components are made with four types of shear wall panels, namely full-size shear wall panel type A (2400 × 1200 mm²), shear wall panel type B (600 × 1200 mm²), shear wall panel type C (400 × 1200 mm²) and shear wall panel type D (800 × 1200 mm²). The form of shear wall construction is made of a stress skin component. Lumber sheathing was designed horizontally (straight sheathing) as a control and diagonally (diagonal sheathing) as a treatment. In designing the shear wall panels, an approximation method is used to determine the forces received by each beam of the slanting body board (18 × 105 mm²) in a 45° direction (Figure 2). The arrangement of the inclined body boards uses the tongue and groove (T and G) system.

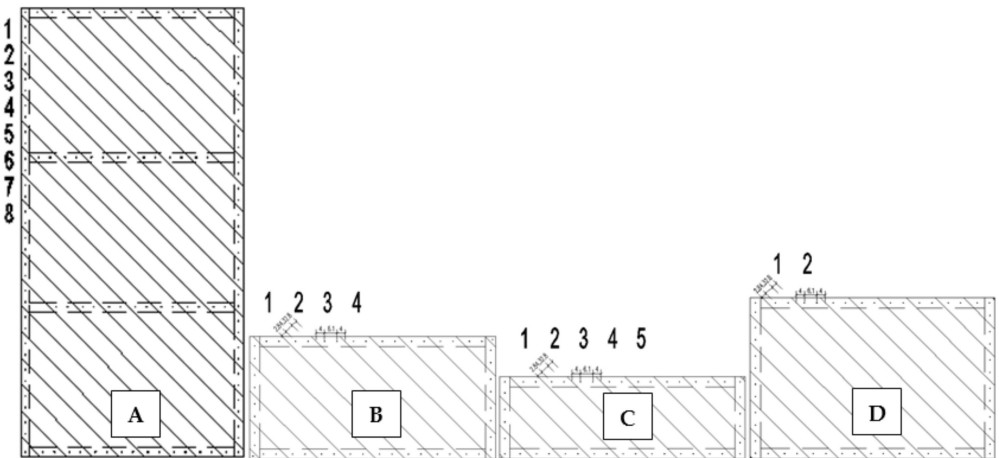

**Figure 2.** CLT-mangium lumber connection forms a shear wall slant body. (**A**) full-size shear wall panel type A; (**B**) shear wall panel type B; (**C**) shear wall panel type C; (**D**) shear wall panel type D.

The normal force that occurs in the shear wall (a pendel) is assumed to be an external force (N). The area of the numbered boards is part of the slanted web, which is assumed to withstand the normal force that occurs in the shear wall panels. The area of the numbered board is the full length of the slanted body board. The nailing distance is shortened by adding rods to the shear wall panel frame.

### 2.3. Manufacturing of Shear Wall Components

The size of the prefab shear wall is 2400 mm × 1200 mm × 86 mm, with several variations in the stress skin model wall design. Making the wall begins with making a frame measuring 50 mm × 70 mm. The frame is then assembled with lumber shearing boards with a T and G pattern measuring 18 mm × 100 mm, and the length varies from 200 mm to 2100 mm. The boards are arranged into a shear wall with the body boards tilted at 45° (diagonal sheathing) using 34/76 (3″ BWG10) 70-mm-long nails. The nailing position is two nails at each end of the panel board, so there are a total of 4 nails per board, which are nailed in pairs. This is intended so that the board cannot rotate. Each frame connection is given one wooden peg at each frame connection between one nail (Figure 3).

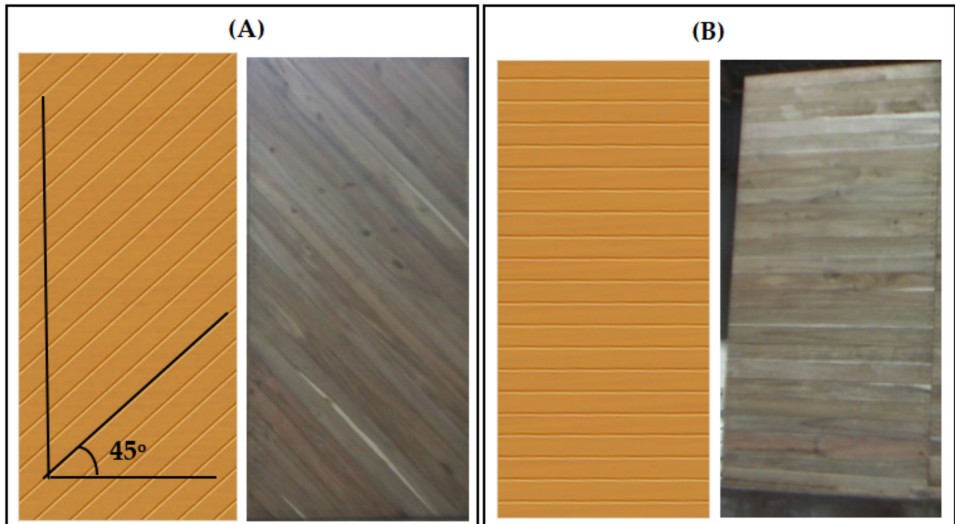

**Figure 3.** Schematic of wall specimens CLT-mangium used as shear wall frames. (**A**) 45° diagonal sheathing wall test specimens; (**B**) Straight sheathing wall test specimens.

The shear wall component consists of four design patterns, including the whole shear wall with the straight shearing pattern as a control (A), the whole shear wall with a diagonal board pattern (diagonal sheathing) angle of 45° (B), the windowed shear wall with a diagonal board pattern (C) and a shear wall with a diagonal board pattern (D), as shown in Figure 4.

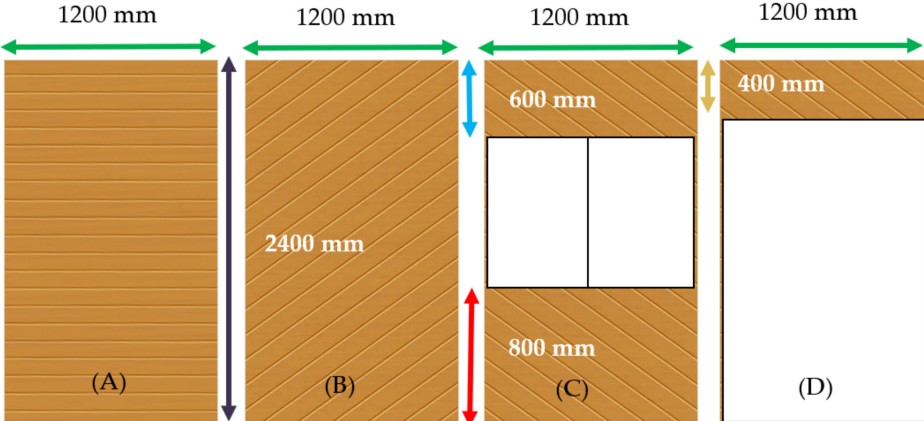

**Figure 4.** (**A**) whole shear wall in board pattern; (**B–D**) shear wall in plain board diagonal, windowed and with doors, respectively.

### 2.4. Assembly of Shear Wall Components

The size of the shear wall test sample is 86 mm × 2400 mm × 2400 mm, in the form of a combination of two shear wall panel components horizontally fastened with bolts. Each shear wall uses the same type of frame, cladding, bolts, nails and nailing pattern. The size and placement of the openings in the form of doors and windows are measured based on the ratio of the area of the covering board (r). Figure 5 shows the schematic drawings of the assemblies and exposed dimensions and describes the location of the openings on each shear wall shape. Shear walls A and B (r = 1.0) have no openings and require capacity measurement under fully enclosed conditions. The ratios of shear walls C–E to shear walls A and B are directly compared for their shear capacity ratios (Figure 6).

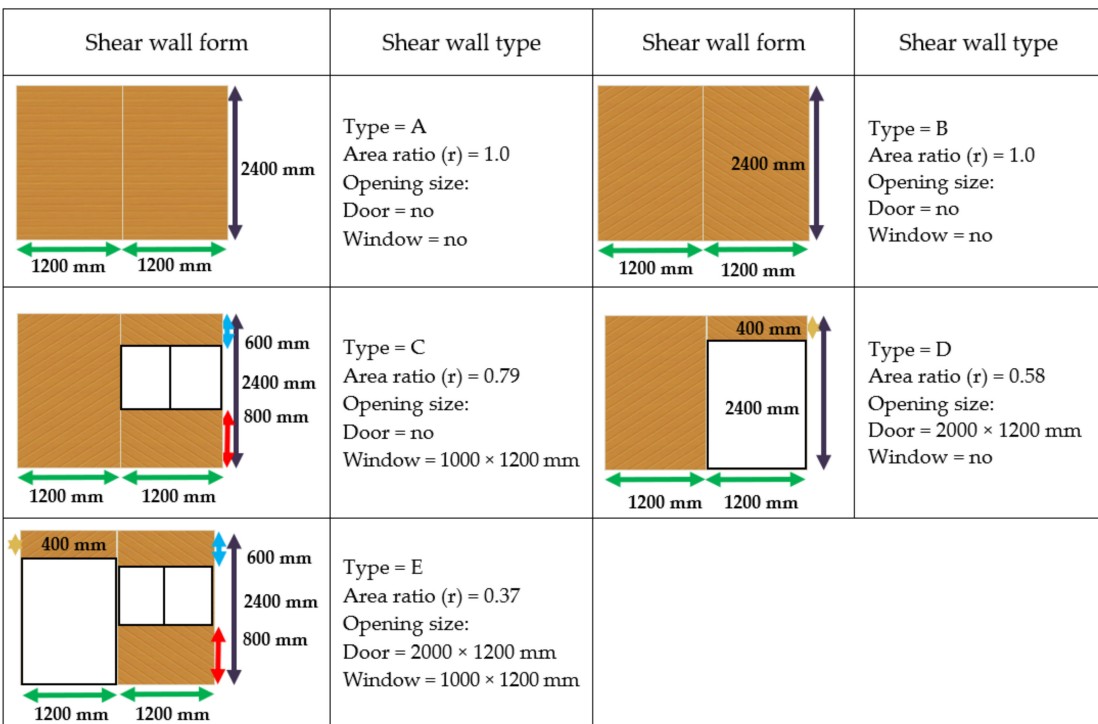

**Figure 5.** Schematic drawing of shear wall components.

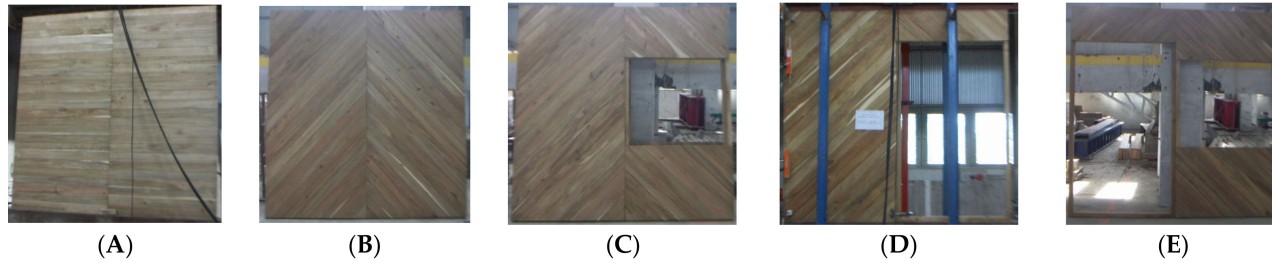

| (A) | (B) | (C) | (D) | (E) |

**Figure 6.** Types of shear wall components as the result of four assembly design patterns. (**A**) Shear wall in board pattern; (**B**) Diagonal sheathing wall; (**C**) Diagonal sheathing wall with windowed; (**D**) Diagonal sheathing wall with door; (**E**) Diagonal sheathing wall with windowed and door.

### 2.5. Testing Shear Wall as Component of Prefabricated Houses Structure

Testing the shear wall components in the form of a horizontal in-plane monotonic load racking stiffness and strength test is based on ISO 22452 [22] for simulating earthquake strength. The loading is given in one direction, namely horizontal loads (racking loads). While the vertical load only functions to withstand the reaction from the racking test so that the value is constant. Tests are required to determine the behavior and reliability of the braces and joints (Figure 7).

The loading is given in the lateral direction and in increments of 0.1 $F_{max,est}$ on the shear wall component. The testing technique on this shear wall is used by adding the load (force) to the displacement (D). The maximum load-estimation value ($F_{max,est}$) is obtained based on the preliminary test. While $F_{max,est}$ is on the test sample that undergoes treatment, which is based on the $F_{max}$ value of the control test sample. If $F_{max,est}$ has been obtained, then the load index given gradually to the horizontal loading is 10% $F_{max,est}$. The determination of $F_{max,est}$ and the load addition index as a treatment depends on the design of the shear wall material (with or without opening), the shear wall material (frame and shearing) and the dimensions of the shear wall. The vertical load (Fv) of 0.4 $F_{max,est}$ is approx. (4 ± 1) mm/min. While the horizontal load (F) is around (2 ± 0.5) mm/min.

Generally, the vertical load (Fv) is more than 25 kN on a sample with a length of 2.4 m. The magnitude of Fv is proportional to the total length of the sample.

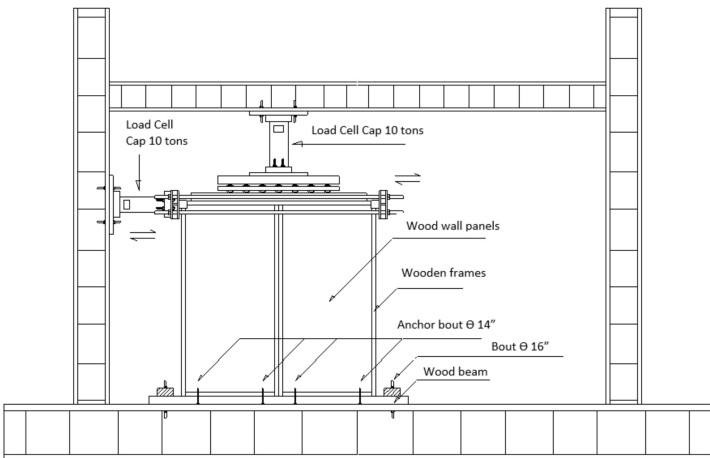

**Figure 7.** Shear wall panel testing settings.

The racking test procedure was carried out in the form of gradually increasing the lateral load by 0.1 $F_{max,est}$ by time, with three steps, namely the stabilizing load cycle is an additional load weighing 0.1 $F_{max,est}$, the stiffness load cycle is an additional load up to a weight of 0.4 $F_{max,est}$, which is carried out in stages in the form of a load of 0.1 $F_{max,est}$, and the strength test adds a load of 0.1 $F_{max,est}$ gradually until the $F_{max}$ of the test object is reached. The loading is stopped if it has exceeded the strength limit of the structure or has exceeded the service limit of the structure in the form of a condition where the board has collapsed or has undergone a deformation/displacement of 100 mm (Figure 8).

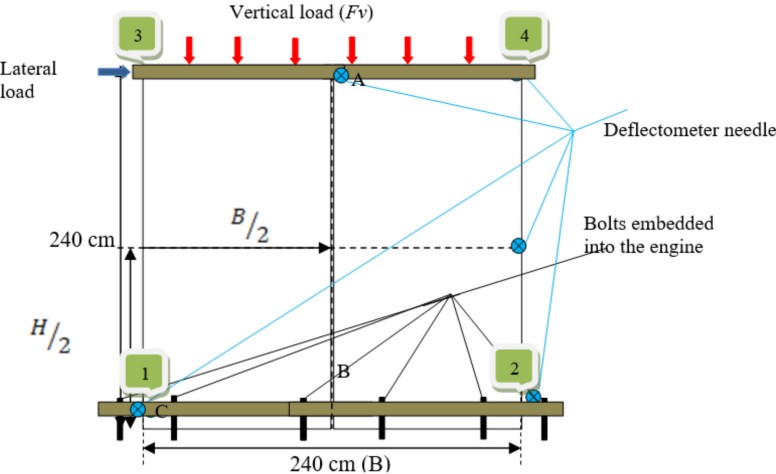

**Figure 8.** Shear wall mechanical strength test by racking test. 1 = Sill plate; 2 = Holddown; 3 and 4 = Steel loading fixture; A = Top plate; B = Bottom plate; C = Anchor bolts.

*2.6. Data Analysis*

Analysis behavior of different opening sizes for each shear wall were used for studying the test results of the CLT-mangium shear wall structural components.

Panel Racking Stiffness (R): Calculated by Equation (1)

$$R = \frac{1}{2}\left[\frac{F_4 - F_1}{\vartheta_4 - \vartheta_1} + \frac{F_{24} - F_{21}}{\vartheta_{24} - \vartheta_{21}}\right] kg \cdot mm^{-1} \tag{1}$$

where: F = applied racking load; $\vartheta$ = the deformation.

Meanwhile, racking strength, which is in the form of the maximum value of the racking load ($F_{max}$), is obtained in the strength test.

The Measurement of Shear Wall Ductility is in the Form of Its Ductility Factor ($\mu$), as in Equation (2)

$$\mu = \frac{\delta_m}{\delta_y}. \tag{2}$$

where: $\mu$ = Structural ducitility factor; $\delta_m$ = The maximum deviation of the structure when it reaches the failure threshold (mm); $\delta_y$ = The structural deviation at the time of the first failure in the structure (mm).

Modulus of Rupture (MOR) and Modulus of Elasticity (MOE) of Shear Wall Components as Cantilever Beams by Equation (3)

$$MOR = \frac{6F_{max}L}{b \cdot h^2} \quad MOE = \frac{4L^3}{b \cdot h^3} \times \frac{F}{\Delta} \tag{3}$$

Earthquake Force with Static Earthquake Analysis Is Equivalent SNI 1726-2002 [23].

The suitability of the earthquake zone is obtained from the value of the horizontal shear force of the earthquake with several assumptions on the characteristics of certain buildings determined by using Equation (4).

$$V = \frac{C_1 I}{R} W_t \tag{4}$$

where: V = Earthquake horizontal shear force (kg); $C_1$ = Earthquake coefficient; I = safety factor of the structure; R = Earthquake reduction factor; $W_t$ = Weight of structure (kg).

## 3. Results and Discussion

### 3.1. Earthquake Resistance on Shearwall Components

3.1.1. Stiffness and Strength Behavior of Shear Wall

The racking stiffness, racking strength, displacement/deformation, comparison of strength and the relative stiffness of various shear wall construction designs diagonal to horizontal shearing on the stress skin component design and shear wall ductility can be seen in Table 1 and Figure 9.

**Table 1.** The stiffness and strength of various shear wall panel component types.

| Types of Shear Wall | Racking Stiffness (R) (kg·mm$^{-1}$) | Racking Strength (kg) | Relative Stiffness | Relative Strength | $\delta y$ (mm) | $\delta m$ (mm) | $\mu$ |
|---|---|---|---|---|---|---|---|
| A | 41 | 216 | 1.00 | 1.00 | 106.58 | 107.58 | 1.01 |
| B | 225 | 486 | 5.48 | 2.29 | 24.19 | 58.19 | 2.41 |
| C | 271 | 505 | 6.60 | 2.34 | 32.99 | 36.19 | 1.10 |
| D | 140 | 356 | 3.41 | 1.67 | 75.99 | 102.28 | 1.35 |
| E1 | 260 | 472 | 6.34 | 2.12 | 94.99 | 95.79 | 1.01 |
| E2 | 11 | 450 | 0.26 | 2.08 | 90.19 | 125.18 | 1.39 |

Note: $\delta y$ = structural deviation at the time of the first failure in the structure (mm); $\delta m$ = Maximum deviation of the structure when it reaches the failure threshold (mm); $\mu$ = structural ductility factor.

Based on Table 1, the shear wall stiffness ranges from 11–271 kg·mm$^{-1}$, where shear wall C has the highest stiffness and shear wall E2 has the lowest stiffness. The relative stiffness ratio of shear wall C was 6.60 times that of shear wall A as a control, and the relative stiffness of shear wall E2 was only 0.26 times that of shear wall A. The shear wall strength ranges from 216–505 kg, where shear wall C has the highest strength and shear wall A has the lowest strength. The relative strength ratio of shear wall C is 2.34 times compared to shear wall A, which has the lowest strength as a control. The vertical joints of the walls with gravity loads showed lower deformation capacity and initial stiffness of the walls when compared to those without gravity loads [24].

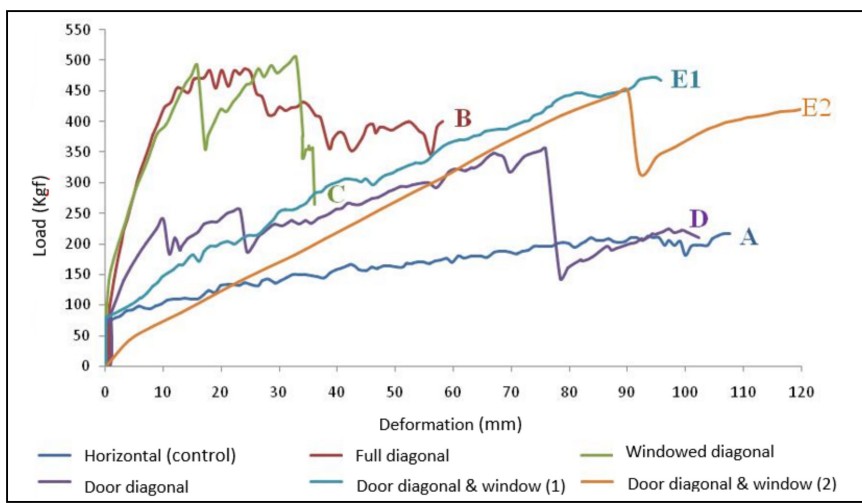

**Figure 9.** Graph of load response comparison—shear wall component deformation.

The lower relative stiffness of the E2 shear wall component is presumably due to the test sample of the shear wall component being tested for the second time. Therefore, this board component panel experienced deformation in the previous test even though its strength is still higher than that of the shear wall A component as a control. This information is useful to determine the strength of building construction against aftershocks, which usually occur after the first earthquake, which is relatively the strongest. Meanwhile, the strength, stiffness and energy dissipation of the single CLT shear walls increased with the increase in the number of connectors [25,26].

The relative stiffness and strength of the shear wall components with the diagonal shearing pattern were stiffer and stronger than the shear wall components with the horizontal shearing pattern used as controls. This is because the construction design of the diagonal shearing shear wall on the design of the stress skin component resists lateral loads stronger and stiffer because it has triangulation properties as well as the nature of the truss compared to the horizontal sheathing design [27].

The shear wall component with the full and windowed diagonal board pattern (C) is stronger than the previous full shear wall component with the diagonal board pattern (B). This is possible because, in the position of the window, the nailing distance is shorter and not only from end to end but also from the end of the side to the middle where there is a window frame [28]. According to Sawata et al. [29], this shear wall component has greater strength because the nailing distance is closer and, therefore, more rigid (functions more as a stiffener/bracing than as a cover/sheathing).

The relatively high stiffness and the strength of the windowed and gated shear wall components (E) are also thought to be due to the tighter spacing between the boards, both on the side frames and in the middle frame. The component panels of this board, in addition to functioning as sheathing, function more as braces even though the sheathing is more open due to the presence of doors and windows [30].

In order for the shear wall components to be stronger and to obtain consistent stiffness and strength values in each form of intact shear wall and opening (windowed and with doors), it is necessary to strengthen the nails in each horizontal frame in the middle of the frame, therefore, this board component panel, in addition to functioning as sheathing, also functions as a brace [31].

### 3.1.2. Construction Failure

Construction failures consist of structural failures where the shear wall component collapses/damages before the deformation reaches 100 mm, and serviceability failures are where the material has not experienced a collapse deformation even though the deformation has reached 100 mm. Based on Table 1 and Figure 9, the test samples for shear wall com-

ponents B (δy = 24.19 mm; δm = 58.19 mm) and shear wall components C (δy = 32.99 mm; δm = 36.19 mm) experienced collapse/damage before the deformation reached 100 mm. In this case, it shows that the two test samples of the shear wall component experienced construction failure in the form of structural failure, which was indicated by a drastic decrease in load bearing after reaching the peak load strength ($F_{max}$).

Construction failure in the form of serviceability failure occurred in three other components, namely shear wall A, the component used as a control, shear wall component D, and shear wall component E (E1; E2). Serviceability failure occurs because the shear wall has very ductile properties, where the components have not experienced deformation/collapse even though the deformation has reached 100 mm (δy; δm = 100 mm). The failure of the structure is not visible; what happens is that the structure experiences a very large horizontal displacement of up to 100 mm, as required in ISO 22452 [22]. The use of controlled ductile rocking shear walls with low-damage connections can be an efficient alternative to a traditional high-damage design in order to mitigate earthquake-induced damage. Similar to the results of the research conducted by Hashemi and Quenneville [32] on a rocking CLT wall with proposed hold-down connectors, have shown potential for use in earthquake-resistant low- to moderate-rise CLT structures.

### 3.1.3. Ductility

Ductility is the ability of a house structure to experience repeated large post-elastic deviation and commute due to earthquake loads at the expense of the earthquake that caused the first collapse while maintaining sufficient strength and rigidity, so the structure of the house remains standing although it has already been destroyed. Shear wall ductility measurements by measuring the ductility factor (μ) show how structure system failures (brittle or ductile) occur and are a parameter for the comparison between the measurement results of the test with the design value. The lowest shear wall ductility factor is 1.01 for the A shear wall, and the highest value is 2.41 for the B shear wall.

The ductility factor of a house (μ) is the ratio between the maximum deviation of the house structure due to the influence of an earthquake plan when it reaches collapse conditions (δm) and the deviation of the house structure at the first collapse δy, namely: $1.0 \leq \mu = \frac{\delta m}{\delta y} \leq \mu m$. If the house ductility structure factor is μ = 1.0, the structure of the house is fully elastic. At full elasticity, the condition of the house structure at collapse condition is achieved at the same time as the first collapse in the structure (δm = δy). Meanwhile, if the level of house structure ductility for the fully elastic structure is 1.0 and the fully ductile house structure is 5.3, the structure of the house will be partially ductile. Additionally, if the level of house ductility, where the structures experience the large post-elastic deflection under collapse conditions, achieving a ductility factor of 5.3, the structure of the house will be fully ductile. According to the test results by Kang et al. [33], when using Glued-Laminated Timber (Glulam) post and beam structures with bolted connections, bolted connections have barely satisfactory lateral performance. Furthermore, the lateral resistance and stiffness of glulam post and beam structures are relatively weak, which are semirigid with low stiffness, ductility and load-carrying capacity.

Based on the description, all partial ductile shear wall types are A, B, C, D, E1 and E2. Shear wall types that behave fully elastic are types A and E1. While during these tests, it is not fully ductile because there is no μ 5.3, and not all types of house system structures can be fully ductile. Partially ductile behavior has been met 1.0 < μ < μm, so the planner or building owner can choose the value of μ based on his desire during the planning of a house structure. Lumber sheathing is extensively used for wood frame shear walls. Lumber sheathing is made horizontally (straight sheathing), and the stiffness and strength are relatively weak and flexible. Meanwhile, in the treatment using a diagonal sheathing type, the stiffness and strength are stronger and stiffer due to the nature of triangulation as well as the nature of the truss.

### 3.1.4. Failure/Damage

We documented the failure modes for the diagonal pattern with horizontal and vertical loading tests were distinguished as the form of a shift between the tongue and groove board panels on the components frame shear walls (Figure 10A) and the detachment of the shear wall frame at the joint points (Figure 10B). Meanwhile, the failure caused by the component shear wall with a diagonal pattern in the form of the gap formation (gap) between the composition of the panel boards diagonally across the lower part due to the imposition of lateral force (Figure 10D) and damage to the structure due to fracture and release of shear wall frame on the amplifier/stiffener (brace) and points connection (Figure 10C). Yang et al. [34] suggested that CLTs are susceptible to producing excessive bending stresses and are prone to failures in tension perpendicular to the grain direction.

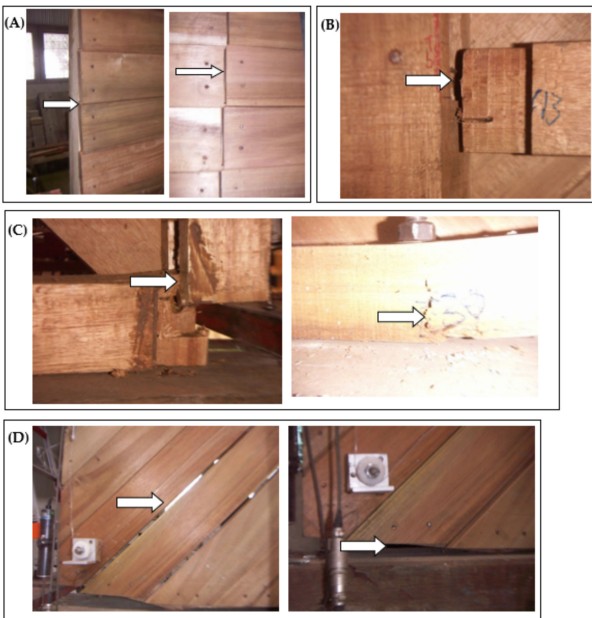

**Figure 10.** Damage that occurs in CLT-mangium shear wall components. (**A**) = Shift between tongue and groove board panels on components frame shear walls lumber sheathing horizontal type (straight sheeting) due to lateral forces; (**B**) = Detachment of the shear wall frame at the joint points; (**C**) = Structural damage due to fracture and detachment of the shear wall frame at the joint points; (**D**) = Damage in the form of the formation of gaps between the arrangement of panels of diagonal boards at the bottom due to the imposition of lateral forces.

Sylvian et al. [35] reported that, when a load is applied, it causes excessive deflection followed by rolling shear failure due to the low out-of-plane shear strength, which is its principal drawback. Furthermore, the withdrawal resistance of stainless-steel nail fasteners loaded perpendicular to the grain (perpendicular to the surface and tangential) had higher withdrawal resistance than those loaded parallel to the grain (edge) [36].

### 3.1.5. The Stiffness and Strength Values of Share Wall Components as Cantilever Beams

If an object undergoes a load test on one side only in the form of a lateral monotonic test and the test object cannot rotate at that point (flops), the test object is called a cantilever beam [24]. Because the left side undergoes a pressure/restraint and the right side is free for deflection, then the test object is called a cantilevered reinforced wooden shear wall component.

According to the results in Table 2, the lowest stiffness (MOE) of the A shear wall component type is only 50.41 MPa, and the highest is evident in type B, with 1418.83 MPa. Highest Stiffness (MOE) on the shear wall component type reaches 28 times the stiffness (MOE) of the elastic shear wall component types. Medium strength (MOR) at the lowest shear wall component in type A is 18 kg/cm$^2$ and the highest on the type C at 41 kg/cm$^2$

can be two to three times stronger than the weakest shear wall panel component type. Wadi et al. [37] suggested that cross-laminated walls without a diagonal strut have approximately double the horizontal strength of the panels. Furthermore, they reported that a diagonal arrangement significantly increases the lateral load resistance of cross-laminated walls, particularly under compression conditions.

**Table 2.** Stiffness (MOE) and strength (MOR) of some shear wall panel components.

| Types of Shear Wall | Pmax | h (mm) | b (mm) | L (mm) | Linear Equation | $R^2$ | MOE (MPa) | MOR (MPa) |
|---|---|---|---|---|---|---|---|---|
| A | 216 | 86 | 2400 | 2400 | y = 1.42x + 91.45 | 0.96 | 514 | 18 |
| B | 486 | 86 | 2400 | 2400 | y = 39.94x + 81.15 | 0.99 | 14,468 | 39 |
| C | 505 | 86 | 2400 | 2400 | y = 26.78x + 140.6 | 1.00 | 9701 | 41 |
| D | 356 | 86 | 2400 | 2400 | y = 18.48x + 74.10 | 0.99 | 6694 | 29 |
| E1 | 472 | 86 | 2400 | 2400 | y = 5.022x + 95 | 0.98 | 1819 | 38 |
| E2 | 450 | 86 | 2400 | 2400 | y = 4.914x + 23.47 | 1.00 | 1780 | 37 |

Note: Pmax = load maximum; h = thickness; b = width; L = length; $R^2$ = Regression coefficient.

The MOE and MOR values are low because the shear wall component, which is considered a cantilever beam, is assumed to be a complete beam, whereas the shear wall component only consists of a frame and a layer of wall cladding derived from boards in the form of lumber sheathing only. Therefore, naturally, the value of stiffness and strength is far below the value of the intact beam.

*3.2. Analysis of the Behavior of Shear Wall Components Due to the Influence of Earthquake Loads*

Based on the calculation of the seismic force of the prefabricated house design from CLT-mangium in six earthquake zones, the total shear force due to the earthquake was obtained. These data become the basis for grouping shear wall component panels in receiving lateral forces, as shown in Table 3.

**Table 3.** Feasibility of shear wall component panels based on the earthquake zone.

| Types of Shear Wall | Load-Shear Wall Component Deformation | | Earthquake Zone | |
|---|---|---|---|---|
| | $P_{max}$ (kg) | Deformation (mm) | | |
| A | 216 | 106.58 | 2 | Low |
| B | 486 | 24.19 | 4 | Moderate |
| C | 505 | 32.99 | 5 | Severe |
| D | 356 | 75.99 | 3 | Moderate |
| E1 | 472 | 94.99 | 4 | Moderate |
| E2 | 450 | 90.19 | 4 | Moderate |

Note: Based on the results of earthquake force calculations by SNI 1726-2002 [23].

Based on the feasibility of shear wall component panels by the earthquake zone in Table 3, the type A design of the CLT-mangium shear wall panel component is suitable for application in low-intensity zones (2), types B, D, E1 and E2 are suitable for moderate intensity zones (3, 4) and type C is suitable for severe intensity zones (5). It was believed that the strength of the horizontal board component design are lower than that of the diagonal board design. Di and Fu [38] reported that the degeneration of the shear wall of CLT stiffness due to earthquake damages would result in decreasing the internal forces acting on the walls while those on the frames increase. They further explained that the shear force and bending moment of the bottom frame columns rise drastically, which may greatly reduce the safety margin and should be considered in practical design. Tang and Zhang [39] studied the seismic performance of shear wall systems and concluded that incorporating a flexible foundation can reduce the damage probability of the shear wall structures. Chang et al. [40] also reported that the energy dissipated at the footing–soil interfaces accounts for a large amount of the dissipated energy of the total system.



## 4. Conclusions

Horizontal-type boards have relatively weaker stiffness, strength and flexibility than the type of diagonal sheathing. Meanwhile, intact shear wall components with diagonal board pattern (B) and intact diagonal board and windowed pattern (C) experienced construction failure in the form of structural failure. Shear wall types (A, D, E1, and E2) experience construction failures in the form of serviceability failures. The damage to the shear wall components of the horizontal board type is in the form of shifting between the boards, while the diagonal-board-type shear wall component is in the form of gaps between the arrangement of the lower diagonal board panels. The design of shear wall panel components from CLT-mangium can be applied to various earthquake intensities (low/2 to high/5).

**Author Contributions:** Conceptualization, R.D. and S.; methodology, R.D. and S.; validation, T.K.; formal analysis, T.K. and R.D.; investigation, R.D. and J.M.; resources, S. and Y.S.; data curation, S. and J.M.; writing—original draft preparation, R.D.; writing—review and editing, R.D. and S.; visualization, R.D. and J.M.; supervision, S.; project administration, R.D.; funding acquisition, R.D. and T.K.; format analysis, Y.S.; Validation and visualization, A. and W.S. All authors have read and agreed to the published version of the manuscript.

**Funding:** This research was funded by Institut Teknologi Bandung (ITB), Indonesia No. 905/IT1.C11/KU/2021, with the Leading Research Development scheme.

**Data Availability Statement:** Not applicable.

**Acknowledgments:** The authors would like to thank the Institut Teknologi Bandung (ITB), Indonesia, for providing Research Grant. The authors would also like to thank the Building Structure and Construction Center, The Settlement Research and Development Center, Minister of Public Works and Human Settlements, Bandung, for providing the necessary facilities for testing.

**Conflicts of Interest:** The authors declare no conflict of interest. The funders had no role in the design of the study; in the collection, analyses, or interpretation of data; in the writing of the manuscript, or in the decision to publish the results.

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
