# Peer review of "Analysis Behavior of Openings on Full-Size Cross-Laminated Timber (CLT) Frame Shear Walls Tested Monotonically"

_forests, doi:10.3390/f14010097_

Round 1
Reviewer 1 Report
Dear Authors
I read the text of the manuscript and made only minor observations. Please pay attention to them in order to further improve the submitted Manuscript.

Author Response
Point 1: Line 15: (largura x comprimento x espressura). ….. That’s it …… Please make it clear in the text.
Response 1: yes we change: 2400 mm x 1200 mm x 68 mm (see highlight green in manuscript)
Point 2: Line 18-19: Do not cite references in the Abstract (even if it is a standard). Say, only that it have been tested according to the specific standard for testing timber structural panels
Response 2: yes we have been deleted (see highlight green in manuscript)
Point 3: Line 57-58: For carbon fiber-reinforced polymers, Husin et al. [13] tested various types of shear walls such as ……
The sentence was a bit meaningless, please check if it got better that way
Response 3: yes we agree (see highlight green in manuscript)
Point 4: Line 79-85: Do not include the methodological part of your work in the objectives. For this there is already the item Materials and Methods. Here you must present a justification for your work, showing what is new in relation to the others, and transcribe here the objectives stated in the abstract.
Response 4: We have been change this sentence to be “This study aims to determine were 1) determine the effects of openings on full-size CLT-mangium frame shear walls tested monotonically, 2) calculating seismic forces to determine the appropriate earthquake zone.” (see highlight green in manuscript)
Point 5: Line 90 and 97: Always use scientific notation: … g.cm-3
Response 5: yes we agree (see highlight green in manuscript)
Point 6: Line 91 and 93: … 1200 mm x 140 mm x 5 mm (length x width x thichness)
Response 6: yes we agree (see highlight green in manuscript)
Point 7: Line 98: Please inform the corresponding value in MPa
Response 7: yes we agree (see highlight green in manuscript)
Point 8: Line 105 and 114: Make it clear in the text, at least once, what dimensions the value (240 x 120 m2, represent
Response 8: We have been change to type A (2400 × 1200 mm2), shear wall panel type B (600 × 1200 mm2), shear wall panel type C (400 × 1200 mm2) and shear wall panel type D (800 × 1200 mm2). (see highlight green in manuscript)
Point 9: Line 124: The same. Check and correct the other cases in the text. Avoid using two units, for example: mm and cm. Try to standardize
Response 9: We have been change to 50 mm x 70 mm. (see highlight green in manuscript)
Point 10: Line 168: Here you have two units (mm and cm). Check and correct
Response 10: We have been change to 86 mm x 2400 mm x 2400 mm (see highlight green in manuscript)
Point 11: Figure 5. Check all other cases and correct when necessary
Response 11: We have been change to all units in figure 5 (see highlight green in manuscript)
Point 12: Line 184: Cite the corresponding reference number, as in other cases and cite correctly in the reference list
Response 12: Yes we done (see highlight green in manuscript)
Point 13: Figure 7: Arrange the figure closest to its first citation in the text
Response 13: Yes we done (see highlight green in manuscript)
Point 14: Figure 8: Always remember that the title of Figure and tables must be self-explanatory and refer to their content. Thus, inform the meaning of the letters and numbers used in the figure
Response 14: Yes we done: Figure 8. Shear wall mechanical strength test by racking test. 1 = Sill plate; 2 = Holddown; 3 and 4 = Steel loading fixture; A = Top plate; B = Bottom plate; C = Anchor bolts (see highlight green in manuscript)
Point 15: Line 246: make clear the meaning of each letter or symbols used in the Equation
Response 15: Yes we done: Where: F = applied racking load; = the deformation (see highlight green in manuscript)
Point 16: Line 255 and 273: Where: µ = Structural ducitility factor; δm = Maximum deviation of the structure when it reaches the failure threshold (mm); δy = Structural deviation at the time of the first failure in the structure (mm). Proceed in the same way with respect to Equation 1
Response 16: Yes we done (see highlight green in manuscript)
Point 17: Line 391: Please write in full, at least the first time this type of panel was cited
Response 17: Yes we done: Glued-Laminated Timber (Glulam) (see highlight green in manuscript)
Point 18: Line 456: Please inform the corresponding value in MPa
Check and do the same for the other cases in text
Response 18: Yes we done: 50.41 MPa (see highlight green in manuscript)
Point 19: Table 2 and 3: Inform the meaning of the letters and symbols at the foot of the table. Always remember that the content presented in tables and figures must be self-explanatory
Response 19: Yes we done: Note: Pmax= load maximum; h = thickness; b = width; L = length; R2 = Regression coefficient (see highlight green in manuscript)
Point 20: Conclusion: Paragraph too long (21 lines). Long paragraph (much more than 10 lines) confuse and distract the reader. As there are Conclusions, I advise you to break them by topics, according to the property/characteristic being evaluated. Do not confuse Conclusions with summary of.
Response 20: Conclusion: Horizontal type boards has relatively weaker stiffness and strength and flexibility than the type of diagonal sheathing. Meanwhile, intact shear wall components with diagonal board pattern (B) and intact diagonal board and windowed pattern (C) experienced construction failure in the form of structural failure. Shear wall types others (A, D, E1, and E2) experience construction failures in the form of serviceability failures. The damage to the shear wall components of the horizontal board type is in the form of shifting between the boards, while diagonal board type shear wall component is in the form of gaps between the arrangement of the lower diagonal board panels. The design of shear wall panel components from CLT-mangium can be applied to various earthquake intensities (low/2 to high/5). (see highlight green in manuscript).
Point 21: References: Mention in the References, also technical norms used in the production of the manuscript. Check that all references are cited correctly, according to the number linked in the text
Response 21: Yes we done
Point 22: References: Whenever possible, cite the DOI of the publication
Response 22: Yes we done. (see highlight green in manuscript).

Reviewer 2 Report
The manuscript is detailed in all itens. Just some minor corrections are necessary.

Author Response
Point 1: Line 57-58: Something wrong in the phrase
Response 2: We change this sentence to : For carbon fiber-reinforced polymers, Husin et al. [13] tested various types of shear walls such as various height, width and openings under monotonic and cyclic load
(see highlight green in manuscript)
Point 2: Line 264: Please add legend, Fmax, b, h …..
Response 2: Where:
Fmax = Load at the cracked point
L = Total length of support span
b = Total width of support span
(see highlight green in manuscript)
